# Pyruvate Dehydrogenase Inhibition Leads to Decreased Glycolysis, Increased Reliance on Gluconeogenesis and Alternative Sources of Acetyl-CoA in Acute Myeloid Leukemia

**DOI:** 10.3390/cancers15020484

**Published:** 2023-01-12

**Authors:** Rebecca Anderson, Kristin M. Pladna, Nathaniel J. Schramm, Frances B. Wheeler, Steven Kridel, Timothy S. Pardee

**Affiliations:** 1Section on Hematology and Oncology, Comprehensive Cancer Center of Wake Forest Baptist Health, Medical Center Boulevard, Winston-Salem, NC 27157, USA; 2Department of Cancer Biology, Comprehensive Cancer Center of Wake Forest Baptist Health, Winston-Salem, NC 27157, USA; 3Cornerstone Pharmaceuticals Inc., Cranbury, NJ 08512, USA

**Keywords:** leukemia, therapy, metabolism, mitochondria

## Abstract

**Simple Summary:**

Acute myeloid leukemia (AML) is an aggressive cancer with poor outcomes that needs new treatments. One new approach to treat AML is to target its metabolism. A large phase III clinical trial using a metabolic inhibitor, devimistat, did not show any benefit for patients. One reason could be that AML cells can change their metabolism in the presence of devimistat. This study looked at how AML cells change their metabolism when devimistat is present. It is hoped that, by understanding how AML cells resist devimistat, new approaches can be developed.

**Abstract:**

Acute myeloid leukemia (AML) is an aggressive disease characterized by poor outcomes and therapy resistance. Devimistat is a novel agent that inhibits pyruvate dehydrogenase complex (PDH). A phase III clinical trial in AML patients combining devimistat and chemotherapy was terminated for futility, suggesting AML cells were able to circumvent the metabolic inhibition of devimistat. The means by which AML cells resist PDH inhibition is unknown. AML cell lines treated with devimistat or deleted for the essential PDH subunit, *PDHA*, showed a decrease in glycolysis and decreased glucose uptake due to a reduction of the glucose transporter GLUT1 and hexokinase II. Both devimistat-treated and *PDHA* knockout cells displayed increased sensitivity to 2-deoxyglucose, demonstrating reliance on residual glycolysis. The rate limiting gluconeogenic enzyme phosphoenolpyruvate carboxykinase 2 (PCK2) was significantly upregulated in devimistat-treated cells, and its inhibition increased sensitivity to devimistat. The gluconeogenic amino acids glutamine and asparagine protected AML cells from devimistat. Non-glycolytic sources of acetyl-CoA were also important with fatty acid oxidation, ATP citrate lyase (ACLY) and acyl-CoA synthetase short chain family member 2 (ACSS2) contributing to resistance. Finally, devimistat reduced fatty acid synthase (FASN) activity. Taken together, this suggests that AML cells compensate for PDH and glycolysis inhibition by gluconeogenesis for maintenance of essential glycolytic intermediates and fatty acid oxidation, ACLY and ACSS2 for non-glycolytic production of acetyl-CoA. Strategies to target these escape pathways should be explored in AML.

## 1. Introduction

Acute myeloid leukemia (AML) is an aggressive disease that is characterized by abnormal proliferation of immature myeloid cells that fail to differentiate. AML is the most common acute leukemia in adults and is a disease of the elderly, with a median age of diagnosis of 68 years old [1]. The 5-year overall survival (OS) for older AML patients is poor and recent improvements in standard therapies have impacted the OS minimally. In a 16-year period, the 5-year survival improved from 19% to 35% in patients younger than 60, while in patients over the age of 60, it remains a dismal 11% [1]. This is felt to be due to a combination of frail patients and resistance to chemotherapy [2].

Resistant leukemia cells show an increased oxidative phosphorylation signature [3]. Several studies have now noted the critical role of mitochondria in cancer cell metabolism and response to therapy [4,5,6,7]. As a result, there are many therapeutics that target different parts of the mitochondria in clinical trials with only limited success [8,9,10]. One such drug is devimistat (CPI-613), a novel therapeutic that targets two key enzyme complexes in the TCA cycle, pyruvate dehydrogenase complex (PDH) and α-ketogluterate dehydrogenase (KGDH, [11,12]). PDH is responsible for the generation of acetyl-CoA from glucose-derived pyruvate. Acetyl-CoA is a central metabolite that ties together multiple anabolic and catabolic pathways and serves as an important post-translational protein modifier (reviewed by Shi and Tu [13]). AML cells increase oxygen consumption in response to chemotherapy and this is a source of resistance [9]. AML cells treated with devimistat or those with genetic deletion of *PDHA* have increased sensitivity to chemotherapeutics [9]. *PDHA*-deleted cells showed significant alterations in mitochondrial function and structure, highlighting the importance of functional PDH and its product, acetyl-CoA [9]. Unfortunately, the phase III trial of the addition of devimistat to chemotherapy for relapsed or refractory AML patients was terminated early for futility. A better understanding of the metabolic adaptations that AML cells undergo when treated with devimistat is needed.

Mitochondrial enzyme mutations occur in tumors. Succinate dehydrogenase (SDH) mutant tumors have shown increased glycolysis in response to defects in their mitochondrial function [14]. Fumarate hydratase (FH) mutant tumors rely on heme oxygenase to partially run the TCA cycle [15]. This demonstrates that cancer cells can compensate for mitochondrial dysfunction by different salvage pathways.

Here we identify mechanisms of AML cell survival in response to PDH inhibition with devimistat or *PDHA* deletion. Glucose uptake and retention decrease in response to devimistat treatment or *PDHA* deletion, while gluconeogenesis increases. The gluconeogenic amino acids glutamine and asparagine are protective against devimistat, and PDH inhibition sensitizes AML cells to asparaginase treatment. Fatty acid oxidation, ATP citrate lyase (ACLY) and acyl-CoA synthetase short chain family member 2 (ACSS2) are alternate sources of acetyl-CoA and contribute to resistance to devimistat. Finally, fatty acid synthase (FASN), a major consumer of acetyl-CoA, is inhibited in response to devimistat.

## 2. Materials and Methods

### 2.1. Cell Culture, Glucose Uptake and Cell Viability

MFL2 and RHRAS cells (MFL2 cells were a kind gift from Dr. Scott Lowe, and RHRAS cells were a kind gift from Dr. Greg Wang) were grown in stem cell medium at 37 °C with 5% CO_2_ as previously described [9,16]. K562, HL60 and OCI cells (obtained from the American Type Culture Collection (ATCC)) were grown in RPMI medium supplemented with FBS, L-glutamine, and Pen-Strep. Glucose uptake was measured by Glucose Uptake-Glo^TM^ (Promega, Madison, WI, USA) according to the manufacturer’s instructions. For viability assays, cells were exposed to the indicated therapy for 72 h and the viable cell population was determined using the Cell Titer Glo assay (Promega, Madison, WI, USA) according to the manufacturer’s protocol.

### 2.2. Cas9 CRISPR Gene Deletion

MFL2 cells were infected with Cas9 expressing vector, MSCV_Cas9_puro (gift from Christopher Vakoc; Addgene plasmid # 65655, Watertown, MA, USA) and infected cells were selected with puromycin. Resistant cells were then transfected with sgRNA expressing vector LRG, :Lenti_sgRNA_EFS_GFP (gift from Christopher Vakoc; Addgene # 65656) tagged with GFP targeting the indicated gene. Gene deletions were confirmed by Western blot. Clonal populations of deleted cells were obtained by serial dilutions.

### 2.3. Quantitative PCR Assays

Total RNA was extracted using RNeasy mini kits as per the manufacturer’s protocol (Qiagen, Valencia, CA, USA), and RNA was converted to cDNA using the iScript cDNA synthesis kit (Bio-Rad, Hercules, CA, USA). Finally, qPCR was performed using the CFX96 thermal cycler using SYBR green Mastermix as per the manufacturer’s protocol (Bio-Rad). Relative mRNA levels were calculated using the ΔΔCT method and normalized to actin. Primer sequences are available on request.

### 2.4. Western Blot

Cell pellets were lysed in Laemmli buffer and quantified by Bio-Rad Protein Assay (Bio-Rad). Then, they were separated by SDS-PAGE and transferred to an Immobilon PVDFmembrane (Millipore, Burlington, MA, USA). Antibodies against PCK (Cell Signaling, Danvers, MA, USA), FASN (BD Biosciences, Franklin Lakes, NJ, USA), ACC (Cell Signaling), P-ACC (Ser79) (Cell Signaling) and β-actin (Abcam, Cambridge, UK) were used. Images of the blots were taken on the Amersham Imager 600. Protein quantification was done using Adobe Photoshop CS6 version 13.0.1.

### 2.5. Amino Acid Supplementation Assay

AML cells were incubated in Hank’s balanced salt solution (HBSS) which contains glucose but no amino acids, and either asparagine or glutamine or both were added. Cells were treated with devimistat for 4 h and returned to complete media, and viability was determined after 72 h using the Cell Titer Glo assay (Promega, Madison, WI, USA) according to the manufacturer’s protocol.

### 2.6. Extracellular Acidification Assays

All acidification rate assays were performed using the XF24 Extracellular Flux Analyzer (Seahorse Bioscience, North Billerica, MA, USA) as per the manufacturer’s instructions. Cells were plated at a density of 600,000 viable cells per well for the acidification rate assay. Data was normalized to viable cell number, and viability was routinely between 96% and 99% as determined by trypan blue exclusion assay (Countess automated cell counter, ThermoFisher, Waltham, MA, USA). The media used for the extracellular acidification rate assay was the glycolysis stress test media, specifically: DMEM with 2 mM L-glutamine—warmed to 37 °C and pH 7.35. Cells were placed in assay media and incubated for 2 h prior to being placed in the assay. For the extracellular acidification rate experiments, glucose was added at a final concentration of 10 mM, oligomycin at 1 μM and 2-deoxyglucose at 100 mM. All assays were done in triplicate (3 wells per condition, each measured in triplicate) and repeated in three independent experiments.

### 2.7. Fatty Acid Synthesis Assay

Fatty acid synthesis activity was determined by incorporation of ^14^C-acetate as described by Bowlby et al. [17]. Briefly, ^14^C-acetate (1 µCi) was added to cells for 2 h. Cells were collected, washed and lysed with hypotonic buffer (20 mM Tris-HCl, 1 mM EDTA, 1 mM DTT, pH 7.5). Lipids were extracted using chloroform/methanol (2:1). Newly synthesized lipids were measured by scintillation counting and counts normalized to total protein.

## 3. Results

### 3.1. PDH Inhibition Leads to Decreased Glucose Uptake and Retention but Reliance on Residual Glycolysis

Deletion of *PDHA* decreased glycolysis in AML cells as measured by the extracellular acidification rate (ECAR) [9]. To confirm and extend this result, K562 and OCI-AML3 cells were treated with devimistat and ECAR was measured. Devimistat decreased ECAR in a dose-dependent manner, confirming that *PDHA* loss or inhibition results in decreased glycolysis in AML cells (Figure 1A,B). To further interrogate this mechanism, glucose uptake in AML cells with deleted *PDHA* was measured. *PDHA*-deleted cells had significantly decreased glucose uptake (Figure 1C). This could be due to reduced glucose uptake or retention or both. To examine glucose uptake, the expression of the high affinity glucose transporter *GLUT1* was assessed. *PDHA* deletion or devimistat treatment resulted in significantly decreased expression of *GLUT1*, suggesting that PDH activity is required for optimal *GLUT1* expression (Figure 1D,E). Hexokinase II (HKII) catalyzes the phosphorylation of glucose, resulting in its sequestration in the cell. It docks to the mitochondrial membrane and can be degraded during mitochondrial stress and turnover [18]. As PDH inhibition or loss results in mitochondrial turnover [8], this could destabilize HKII. To test this, devimistat-treated cells and *PDHA*-deleted cells were treated with the proteasome inhibitor bortezomib. Devimistat-treated cells with intact PDH (Figure 1F) and *PDHA*-deleted untreated cells (Figure 1G) showed decreased amounts of HKII, which was rescued with bortezomib. To determine if *HKII* gene expression is affected by PDH loss or inhibition, qPCR was performed. *HKII* expression was reduced in devimistat-treated cells (Figure 1H) but not in the PDH-deleted cells (Figure 1I), possibly reflecting a difference in acute PDH inhibition with devimistat versus chronic loss in *PDHA*-deleted cells. Finally, given the decreased ECAR, the expression of lactate dehydrogenase (*LDHA*) that converts pyruvate to lactate driving ECAR was assessed. There was a decrease in *LDHA* gene expression in both *PDHA*-deleted (Figure 1J) and devimistat-treated cells (Figure 1K). To assess if the residual glycolytic activity is essential, *PDHA*-deleted and devimistat-treated cells were treated with the glycolytic inhibitor 2-DG. Under either condition, the AML cells were highly sensitive to glycolytic inhibition (Appendix A). Taken together, these data show that genetic or pharmacologic inhibition of PDH decreases glycolysis by decreasing glucose import and glucose retention, but AML cells are still reliant on it.

### 3.2. Gluconeogenesis Is a Source of Resistance to Devimistat

PDH inhibition leads to decreased glucose uptake, retention and glycolysis, likely making essential glycolytic intermediates limiting. During glucose deprivation, cancer cells can exhibit an abbreviated form of gluconeogenesis [19]. This pathway is an alternative source of essential glycolytic intermediates in cells with diminished glycolysis. The rate-limiting step of gluconeogenesis is catalyzed by phosphoenolpyruvate carboxykinase (PCK), which converts oxaloacetate to phosphoenolpyruvate. We looked at the mitochondrial isoform, *PCK2*, and found that gene expression was increased in AML cells in a dose-dependent manner when treated with devimistat (Figure 2A); this was correlated with protein levels (Figure 2B,C). To determine if PCK2 activity contributed to resistance, MFL2 cells were treated with devimistat with and without the PCK2 inhibitor 3-mercaptopicolinic acid (3-MPA). PCK2 inhibition significantly sensitized AML cells to devimistat (Figure 2D), indicating that gluconeogenesis is a source of resistance to TCA cycle inhibition. Consistent with increased gluconeogenesis, the mRNA of the gluconeogenic enzyme, *FBP1,* was dramatically upregulated by devimistat (Figure 2E). These data suggest that devimistat treatment results in activation of gluconeogenesis.

### 3.3. TCA Cycle Inhibition Induces Increased Glutamine and Asparagine Dependence

Cancer cells can switch their fuel sources based on nutrient availability [20]. The data above demonstrated that gluconeogenesis is a source of resistance to devimistat. This implies that gluconeogenic amino acids could protect AML cells from devimistat treatment. To confirm this, the gluconeogenic amino acids glutamine and asparagine were tested for their ability to protect AML cells. Glutamine or asparagine rescued the devimistat-induced reduction of MFL2 and RHRAS cell viability, although the combination of the two amino acids did not increase the level of rescue (Figure 3A,B). To further evaluate the dependence on asparagine, AML cells were treated with asparaginase with concurrent devimistat treatment or *PDHA* deletion. *PDHA*-deleted (Figure 3C) as well as devimistat-treated (Figure 3D) AML cells showed increased sensitivity to asparaginase treatment. These data suggest that devimistat engenders an increased reliance on the gluconeogenic amino acids glutamine and asparagine.

### 3.4. Acetyl-CoA Metabolism Is Altered by PDH Inhibition

In addition to the effects seen on glycolysis and gluconeogenesis, PDH inhibition will decrease cellular acetyl-CoA pools. While PDH is a major supplier of cellular acetyl-CoA, a number of other metabolic pathways contribute to acetyl-CoA levels. The mitochondria oxidize fatty acids producing acetyl-CoA, which can be metabolized in the mitochondria or converted to citrate and exported for generating cytosolic acetyl-CoA through ACLY. Therefore, we examined the contribution of fatty acid oxidation to AML cell response to PDH inhibition. Etomoxir inhibits carnitine palmitoyltransferase I (CPT1), which is responsible for transport of fatty acyl chains from the cytosol into the mitochondria for oxidation. AML cells were treated with etomoxir in the presence and absence of devimistat, and viability was assessed. Etomoxir caused a significantly increased sensitivity to devimistat (Figure 4A), suggesting devimistat engenders an increased reliance on fatty acid oxidation. Cellular acetyl-CoA pools are depleted by fatty acid synthesis [13]. Fatty acid synthase (FASN) activity was monitored by C^14^ acetate incorporation and was found significantly decreased in K562 and HL60 cells when treated with devimistat (Figure 4B). ACLY catalyzes the conversion of mitochondria-derived citrate into oxaloacetate and acetyl-CoA in an ATP-dependent manner (reviewed by Icard et. al. [21]). It is localized in both the cytoplasm and the nucleus [22]. In the cytoplasm, it produces acetyl-CoA for fatty acid and cholesterol synthesis [23]. In the nucleus, it provides acetyl-CoA for histone acetylation to maintain expression of genes including *HKII* and *LDHA* [22]. ACLY is therefore a critical node that links citrate availability to changes in expression of genes involved in glucose metabolism (reviewed by Granchi [24]). To assess the role of ACLY in maintaining histone acetylation following devimistat treatment, MFL2 cells were treated with devimistat with and without the ACLY inhibitor NDI-091143 (NDI) for 24 h, harvested and blotted for acetyl-histone H3 lysine 27 (H3L27). Devimistat did not decrease H3L27 acetylation, while the combination of devimistat and NDI did (Figure 4C). To determine the effect of NDI on resistance to devimistat, MFL2 and RHRAS cells were treated with either devimistat or NDI or both for 72 h and viability was assessed. NDI sensitized both MFL2 and RHRAS AML cells to devimistat (Figure 4D). An additional source of acetyl-CoA is ACSS2. ACSS2 catalyzes the ATP-dependent synthesis of acetyl-CoA from acetate. It is important in the maintenance of acetyl-CoA pools, especially in times of glucose scarcity, and maintains tumor growth in low serum conditions [25]. ACSS2 is translocated into the nucleus following glucose depravation [26]. In the nucleus, it produces acetyl-CoA for histone acetylation, which promotes transcription of lysosomal and autophagic genes. Given the central role of ACSS2 in maintaining histone acetylation and expression of genes involved in autophagy under glucose-limiting conditions, it is a likely source of resistance to devimistat. AML cells treated with devimistat with and without an ACSS2 inhibitor for 72 h were assessed for viability. Treatment of AML cells with the ACSS2 inhibitor induced significant sensitivity to devimistat, consistent with the possibility that ACSS2 is a source of resistance (Figure 4E). These data demonstrate that acetyl-CoA metabolism is altered in AML cells following PDH inhibition; these alterations include an increased reliance on non-PDH sources including fatty acid oxidation, ACLY and ACSS2 activity. The metabolic alterations found in response to PDH loss or inhibition are summarized in Figure 5.

## 4. Discussion

Cancer cells adapt to changes in nutrient availability or mutations in metabolic enzymes to gain a survival advantage. One example is that tumors that have defective SDH upregulate glycolysis in order to meet their energetic needs [14]. In contrast, our results suggest that PDH inhibition with devimistat decreases glycolysis by destabilizing HKII and decreasing *GLUT1* expression. This results in the elimination of glycolysis as an escape mechanism for PDH inhibition. Despite this, the residual glycolysis in devimistat-treated cells remains essential, as demonstrated by the enhanced sensitivity of devimistat-treated cells to the glycolysis inhibitor 2-DG. In addition to residual glycolytic activity, devimistat-treated AML cells rely on gluconeogenesis to supply the needed glycolytic intermediates. Gluconeogenesis is indeed the reverse pathway of glycolysis and can generate all glycolytic intermediates. PCK isoforms catalyze the conversion of oxaloacetate to phosphoenolpyruvate, the rate-limiting step of gluconeogenesis. PCK has been shown to be upregulated in multiple cancers and used to adapt to glucose deprivation [19,27,28]. The finding that PCK2 activity is a source of resistance to devimistat in AML cells supports gluconeogenesis as an escape mechanism. This activity of PCK has not been previously described in AML to our knowledge.

Certain amino acids can serve as initial substrates for gluconeogenesis. These include glutamine and asparagine. Both glutamine and asparagine were able to partially protect AML cells from devimistat in a minimal glucose-containing media. The finding that glutamine can protect against TCA cycle inhibition is not surprising, given the established role of glutamine metabolism in leukemia cell survival and resistance [29,30]. Indeed, glutamine has been shown to supplement the needed glycolytic intermediates through a truncated version of gluconeogenesis [20]. The ability of asparagine to rescue is surprising and suggests that it is converted to oxaloacetate to serve as a substrate for PCK2. If true, this would mean that the asparaginase II pathway, consisting of asparagine transaminase and ω-amidase (as reviewed by Cooper et al. [31]), is active. This pathway is not known to be operative in AML. The importance of asparagine was further supported by the increased sensitivity to asparaginase shown by devimistat-treated AML cells. This is complicated by the fact that asparaginase has glutaminase activity [32], but it does establish that the other gluconeogenic amino acids present in the media were unable to compensate for the loss of glutamine and asparagine. This suggests that AML cells have a dependency on asparagine and glutamine in the setting of PDH inhibition.

Apart from essential glycolytic intermediates, acetyl-CoA itself is an essential cellular metabolite [13]. Given that PDH inhibition via devimistat treatment or *PDHA* deletion would deplete acetyl-CoA, it follows that PDH-independent sources would be important for resistance in AML cells. Fatty acid metabolism is an important regulator of the levels of acetyl-CoA, as it is produced by fatty acid oxidation and consumed by fatty acid synthesis. AML cells treated with devimistat displayed an increased reliance on fatty acid oxidation, demonstrating it is a source of resistance. Additionally, fatty acid synthesis via FASN activity was significantly decreased by devimistat, demonstrating a response to acetyl-CoA scarcity. This is consistent with previous findings that devimistat activates adenosine monophosphate activated protein kinase (AMPK [9]). AMPK is a major metabolic regulator, and when activated inhibits FASN activity by phosphorylating and inhibiting acetyl-CoA carboxylase. Additionally, ACLY is an important alternative source of acetyl-CoA for both macromolecule synthesis and protein acetylation [24]. ACLY inhibition led to increased sensitivity to devimistat, demonstrating its role in resistance. Further, combined treatment with ACYL inhibition and devimistat resulted in decreased histone acetylation, suggesting that resulting acetyl-CoA levels are insufficient for epigenome maintenance. Finally, ACSS2 also produces acetyl-CoA independently of PDH. As was the case with ACLY, ACSS2 activity was a source of resistance to devimistat in AML cells, further highlighting the importance of alternative acetyl-CoA sources in the face of TCA cycle inhibition.

## 5. Conclusions

This study demonstrates that PDH inhibition in AML cells with devimistat results in decreased glycolysis and increased reliance on glutamine and asparagine, while upregulating PCK2. In addition, AML cells turn to rely on alternative sources of acetyl-CoA, including ACLY, ACSS2 and fatty acid oxidation, while inhibiting FASN. AML cells demonstrate remarkable metabolic flexibility when challenged with PDH inhibition. This may explain the difficulties in developing clinically active metabolic inhibitors in AML and other cancers. Future efforts should concentrate on rational combinatorial therapies to prevent metabolic escape.

## Figures and Tables

**Figure 1 cancers-15-00484-f001:**
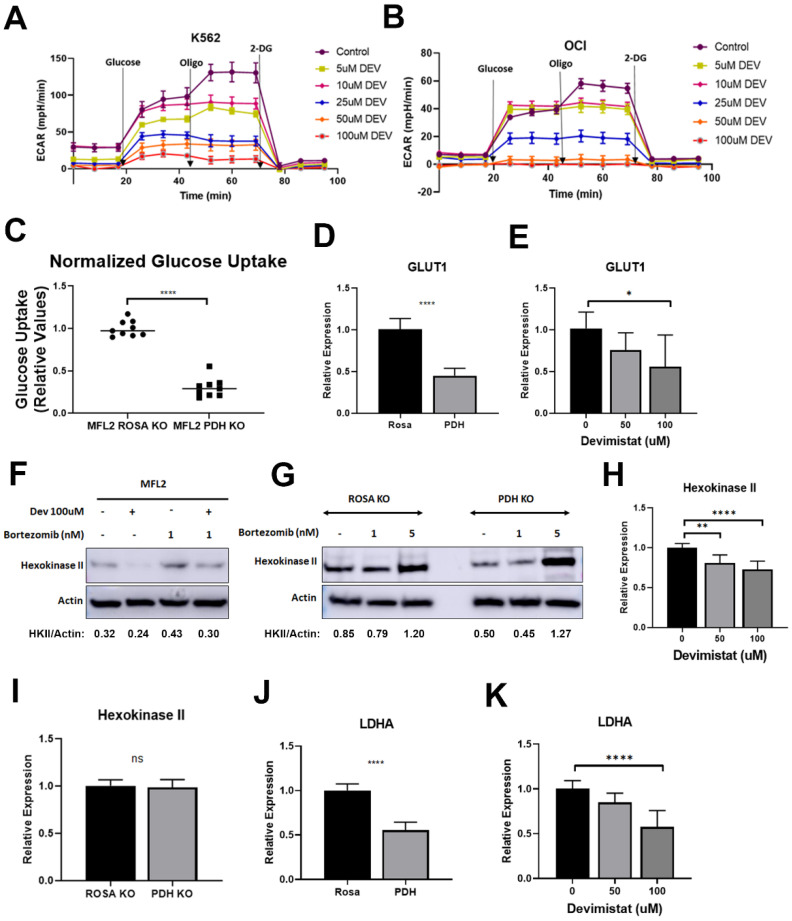
TCA cycle inhibition leads to decreased glycolysis glucose uptake and glucose retention. (**A**,**B**) Extracellular acidification rates (ECAR) following the indicated concentrations of devimistat on K662 (**A**) and OCI-AML3 (**B**) cells. Results are shown of three independent experiments, each done in triplicate. (**C**) Glucose uptake was assessed in ROSA control and PDH-deleted cells. The resulting data were normalized to ROSA control. (**D**,**E**) QPCR of *GLUT1* in *PDHA*-deleted and control ROSA26-deleted cells, and (**D**) MFL2 cells treated with the indicated amount of devimistat for 24 h (**E**). (**F**) Western blot of MFL2 cells treated with devimistat at 100 μM and/or bortezomib at 1 nM for 24 h. Relative densitometry readings below actin. (**G**) Western blot of ROSA and PDH KO cells treated with bortezomib at 1 or 5 nM for 24 h. Relative densitometry readings below actin. (**H**) QPCR analysis of Hexokinase II in MFL2 cells treated with the indicated amount of devimistat for 24 h. (**I**) QPCR analysis of Hexokinase II in ROSA control and PDH-deleted cells. (**J**) QPCR analysis of LDHA in ROSA control and PDH-deleted cells, and (**K**) MFL2 cells treated with the indicated amount of devimistat for 24 h. **** = *p*-value less than or equal to 0.0001; ** = *p*-value less than or equal to 0.01; * = *p*-value less than or equal to 0.05; ns = nonsignificant. The uncropped blots are shown in Appendix A.

**Figure 2 cancers-15-00484-f002:**
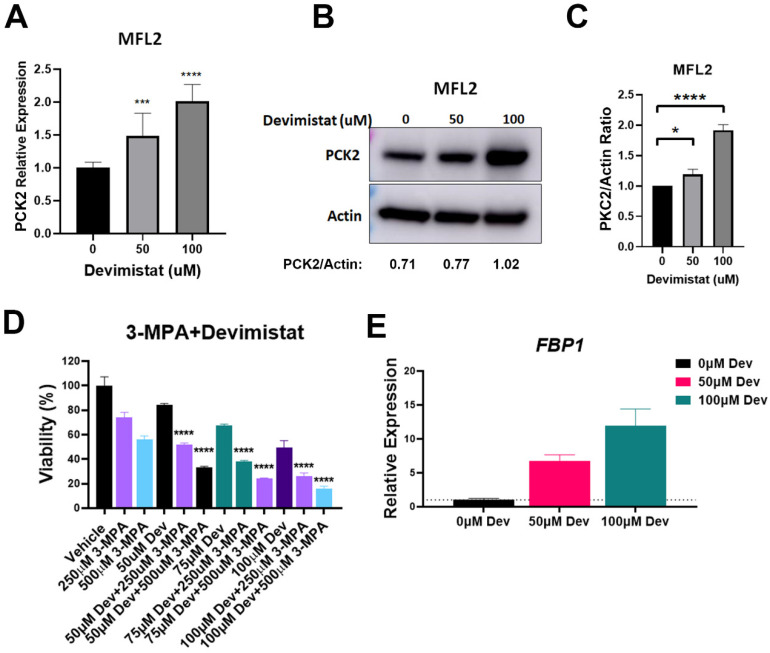
Devimistat induces gluconeogenesis. (**A**) QPCR analysis of PCK2 in MFL2 cells treated with the indicated doses of devimistat for 24 h. (**B**) Western blot analysis of MFL2 cells treated with devimistat at the indicated doses for 24 h. Relative densitometry readings below actin. (**C**) Protein quantification of (**B**) based on actin expression using ImageJ software. (**D**) MFL2 cells were treated as indicated for 72 h and viability assessed. (**E**) MFL2 cells were treated as indicated for 24 h and *FBP1* message levels determined by RT-QPCR. **** = *p*-value less than or equal to 0.0001; *** = *p*-value less than or equal to 0.001; * = *p*-value less than or equal to 0.05. Dev = devimistat, 3-MPA = 3-mercaptopicolinic acid. The uncropped blots are shown in Appendix A.

**Figure 3 cancers-15-00484-f003:**
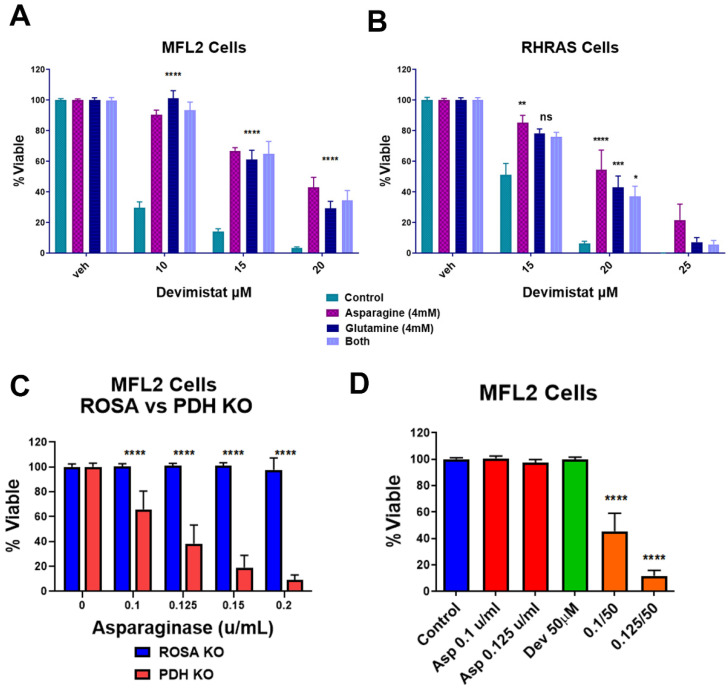
Asparagine and glutamine rescue AML cells from devimistat treatment. (**A**) MFL2 cells were incubated for 4 h in HBSS with 4 mM asparagine, 4 mM glutamine or both, with the indicated doses of devimistat. The cells were then washed and returned to complete media for 72 h and viability was assessed. (**B**) RHRAS cells were incubated for 4 h in HBSS with 4 mM asparagine, 4 mM glutamine or both, with the indicated doses of devimistat. The cells were then washed and returned to complete media for 72 h and viability was assessed. (**C**) ROSA and PDH KO MFL2 cells were incubated with the indicated concentration of asparaginase for 72 h and assessed for viability. (**D**) MFL2 cells were incubated with the indicated concentrations of devimistat, asparaginase or both for 72 h and assessed for viability. **** = *p*-value less than or equal to 0.0001; *** = *p*-value less than or equal to 0.001; ** = *p*-value less than or equal to 0.01; * = *p*-value less than or equal to 0.05. ns = nonsignificant.

**Figure 4 cancers-15-00484-f004:**
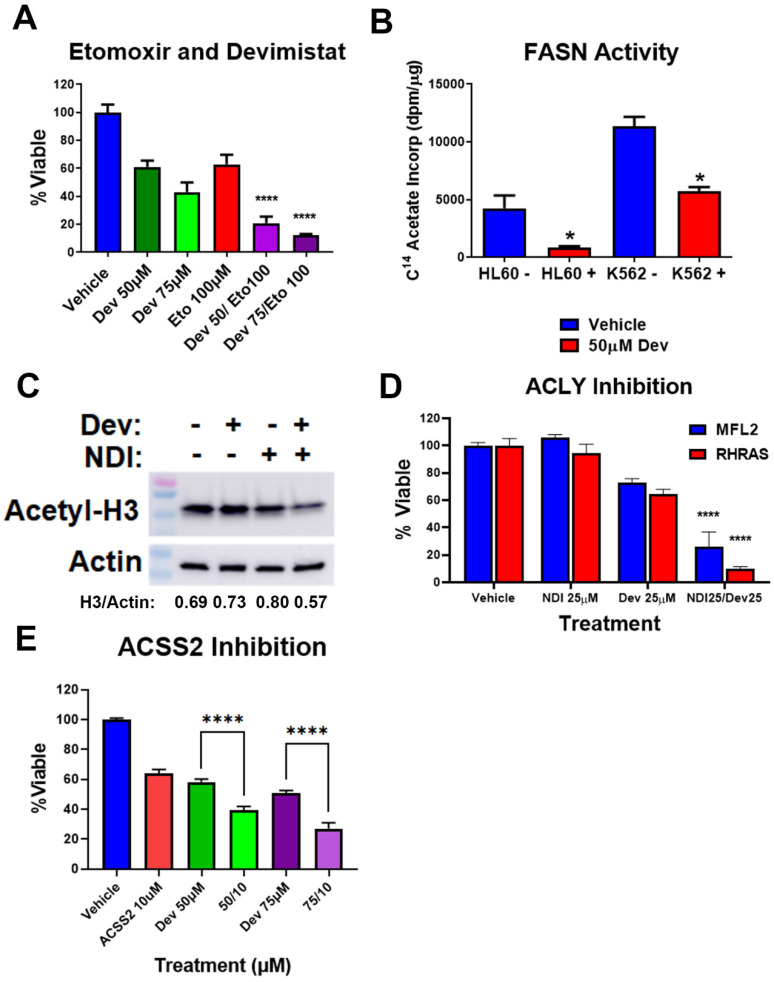
Devimistat alters acetyl-CoA metabolism. (**A**) MFL2 cells were treated with devimistat and etomoxir as indicated for 72 h and viability assessed. (**B**) HL60 and K562 cells were treated with the indicated amount of devimistat for 24 h and FASN activity was assessed. (**C**) MFL2 cells were treated with 100 μM devimistat or 25 μM of the ACLY inhibitor NDI-091143 or both for 24 h, harvested and blotted for acetyl-H3 lysine 27. Actin was used as a loading control. Relative densitometry readings below actin. (**D**) MFL2 and RHRAS cells were treated with devimistat with and without NDI-091143 for 72 h and viability was assessed. (**E**) MFL2 cells were treated as indicated for 72 h and viability was assessed. Dev = Devimistat, Eto = Etomoxir, NDI = NDI-091143, ACSS2 I = ACSS2 inhibitor, **** = *p* < 0.0001, * = *p* < 0.05. The uncropped blots are shown in Appendix A.

**Figure 5 cancers-15-00484-f005:**
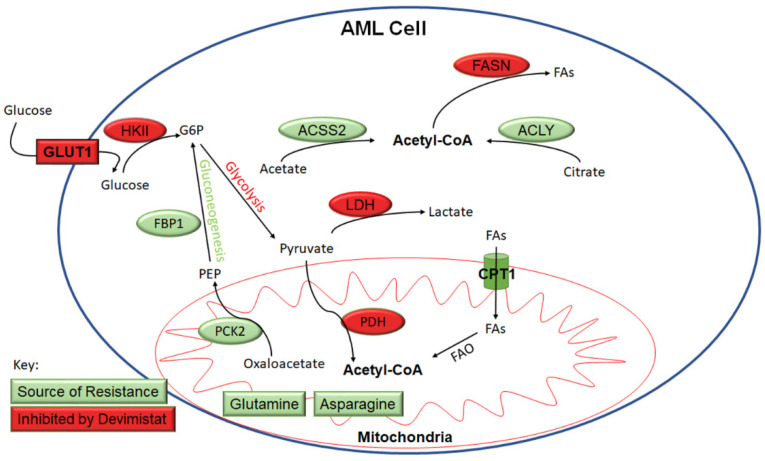
Model of inhibition and AML cell response to devimistat. Shown are the enzymes, metabolites and pathways that are sources of resistance (green) or inhibited by devimistat (red). FAs = fatty acids, FAO = fatty acid oxidation, G6P = glucose-6-phosphate.

## Data Availability

The data presented in this study are available on request from the corresponding author.

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
