# Peer review of "Pyruvate Dehydrogenase Inhibition Leads to Decreased Glycolysis, Increased Reliance on Gluconeogenesis and Alternative Sources of Acetyl-CoA in Acute Myeloid Leukemia"

_cancers, 2023, doi:10.3390/cancers15020484_

Round 1

Reviewer 1 Report

This manuscript entitled: "Pyruvate Dehydrogenase Inhibition Leads to Decreased Glycolysis, Increased Reliance on Gluconeogenesis and Alternative Sources of Acetyl-CoA in Acute Myeloid Leukemia" is perfectly written and makes it possible to answer a concrete problem of the failure of the clinical trial (failure of the use of inhibition of PDH in AML). The results of the various experiments are very clearly presented and the techniques make it possible to respond to the hypotheses tested. This work is suitable for publication.

Author Response

Reviewer 1 comments:

This manuscript entitled: "Pyruvate Dehydrogenase Inhibition Leads to Decreased Glycolysis, Increased Reliance on Gluconeogenesis and Alternative Sources of Acetyl-CoA in Acute Myeloid Leukemia" is perfectly written and makes it possible to answer a concrete problem of the failure of the clinical trial (failure of the use of inhibition of PDH in AML). The results of the various experiments are very clearly presented, and the techniques make it possible to respond to the hypotheses tested. This work is suitable for publication.

We thank the reviewer for their kind words.

Reviewer 2 Report

GENERAL

The paper is well written, in a very plain and easily understandable fashion. The text flows in a very straightforward way, with the relationships between consequential concepts kept extremely simple and clear. There are, however, many minor syntax mistakes. The results obtained are very interesting and clearly described.

As far as the writing is concerned, the Authors should note that the attribute of an adjective should be connected to the adjective itself with a dash. Otherwise, the attribute can look like the subject of sentence, in case the adjective derives from a verb. For example “PDHA deleted”, “devimistat treated”, “glucose containing” should be all connected by a dash. However, this does not apply, for example, to “uniformly labelled”, because an adverb cannot be the subject of a verb; therefore, in this case the dash is not needed. Suggestions about how the text can be improved are listed hereafter

ABSTRACT

Several acronyms are defined, several ar not. The Authors should choose either criterion and adopt it thoroughly.

Likewise, in the Text, for example, PDH is not defined as acronym of pyruvate dehydrogenase.

MATERIALS & METHODS

Space between paragraphs is missing in Page 2. The title Fatty Acid Synthesis Assay should be closer to the paragraph it refers to (Page 3).

Paragraph “Cell culture, glucose uptake, and viabilities” – In the title, change “viabilities” to “cell viability”. In the text, mention to HL60 and RHras cells is missing (I believe that the latter should be written RHras rather than RHRAS).

Lines 73 and 74 – Change “media” to “medium”.

Line 102 – The title “Addback Assay” is kind of awkward.

Line 104 – Delete “added either” and add “were added” right before the full stop.

RESULTS

Lines 143-144 – The concepts of “mitochondrial stress” and “mitochondrial turnover” should be probably linked to each other.

Line 144 – Change “could destabilized” to “could destabilize”.

 Lines 145-148 – Apparently, HKII was suppressed by both transcriptional and post-translational mechanisms in devimistat-treated cells, while by post-translational mechanisms only in PDHA-deleted cells (however, it appears that bortezomib-induced post-translational suppression accounts for most of the observed effect). The transcriptional effects of devimastat are also shown, for example, in Figure 2A. The Authors should address this point.

Figure 1G – It should be specified whether cells are treated with devimastat or not.

Line 156 – The paragraph title should be placed immediately before line 172.

Line 178 – Change “manor” to “manner”.

Figure 2A - Change “PCK2” to “MFL2” (by analogy with Fig. 2B and 2C) and add “PCK2” to “Relative Expression” (ordinate).

Figure 2C – Change “Relative Ratio of PCK2/Actin” to “PCK2/Actin Ratio”.

Line 180 – Add a comma after “resistance”.

Line 180 – “several AML cell lines” seems actually to refer to several subclones of genetically manipulated MFL2 cells. The point should be clarified here as well as in Figure 2D and in the legend to Figure 2D.

Lines 181 and 267– Change “CPI-613” to “devimistat”.  

Line 183 – Change “mRNA of the gluconeogenic enzyme, FBP1” to “the mRNA of the gluconeogenic enzyme FBP1” (add “the”, remove comma).

Line 195 – Change “the nutrient” to “nutrient”.

Line 196 – Resistance: to what should be specified. Probably the two periods of Lines 195-197 should be combined. 

Lines 198-200 - Probably the two periods should be combined to state that “Glutamine or asparagine rescued the devimistat-induced reduction of MFL2 and RHRAS cell viability, although the combination of the two aminoacids did not increase the level of rescue (Figure 3A,B)”.

Figure 3C,D - Here again, like in the case of Figure 2A-C, panel headings are misleading: Rosa vs PDH k/o cells are also MFL2 cells. Furthermore, in Figure 3D, the indications relative to the abscissa are incomplete, while too many colours are used for the bars. Colours should be four, maximum. I would suggest: Control, Asp (same colour, 2 different concentrations), Dev alone, Dev + Asp (same colour, 2 different concentrations of Asp).

Line 219 – Add a comma after “gluconeogenesis”.

Line 220 – Change “acetyl-CoA an” to “acetyl-CoA, a”.

Line 222 – Change “acetyl-CoA for use in the mitochondria or it can be converted” to “acetyl-CoA, which can be metabolized in the mitochondria or converted”.

Line 225 – Change “Carnitine” to “carnitine”.

Line 227 – Change “end” to “or”. Line 228 – Change “significant increased” to “significantly increased” or “significant increase of”.

Line 229 – Change “reliance of” to “reliance on” or “reliance of cells on”.

Line 231 – Change “and was” to “and found”.

Line 233 – Change “mitochondrial derived” to “mitochondrion-derived” or “mitochondria-derived”.

Line 242 – Change “acetylation however” to “acetylation, while” or “acetylation, whereas” or “acetylation, but”

Line 248 – Add commas after “pools” and after “scarcity”.

Line 251 – Add comma after “acetylation”.

Line 256 – Add comma after “devimistat”. Change “it being” to “the possibility that ACSS be”.

Line 258 – Change “inhibition including” to “inhibition; these alterations include”.

Line 259 – Change “A summary of the” to “The”.

Line 260 – Change “is” to “are”. 

Figure 4B – The indications “Vehicle / 50 mM Dev” should be closer to the graph of Figure 4B than to that of Figure 4D.

Figure 4C – This sub-figure is the only one without a title; I would suggest “H3L27 acetylation”. Legend to Figure 5 – “See text for the other abbreviations” should be added at the end of the legend.

DISCUSSION

Line 284 – Change “to supplement needed” to “to supply the needed”.

Line 285 – Add “indeed” after “is”.

Line 299 – Add a comma after “true”.

Line 310 – Add a comma after “acetyl-CoA”.

Lines 311-312 – Change “acetyl-CoA levels as it” to “the levels of acetyl-CoA, which”.

Line 313 – See comment to Line 229.

Line 314 – Change “as” to “is”.

Lines 315 and 321 – Add a comma after “devimistat”.

Line 323 – Add a comma after “acetylation”.

Line 325 – Add a comma after “cells”.

Line 330 – Change “to alternative” to “to rely on alternative”.

REFERENCES

Reference numbers are repeated twice

Author Response

We have addressed all of reviewer 2s comments. 

Reviewer 2 comments:

The paper is well written, in a very plain and easily understandable fashion. The text flows in a very straightforward way, with the relationships between consequential concepts kept extremely simple and clear. There are, however, many minor syntax mistakes. The results obtained are very interesting and clearly described.

We thank the reviewer for their kind words. 

As far as the writing is concerned, the Authors should note that the attribute of an adjective should be connected to the adjective itself with a dash. Otherwise, the attribute can look like the subject of sentence, in case the adjective derives from a verb. For example “PDHA deleted”, “devimistat treated”, “glucose containing” should be all connected by a dash. However, this does not apply, for example, to “uniformly labelled”, because an adverb cannot be the subject of a verb; therefore, in this case the dash is not needed. Suggestions about how the text can be improved are listed hereafter

We thank the reviewer for their thorough review and have made the changes as suggested.

ABSTRACT

Several acronyms are defined, several ar not. The Authors should choose either criterion and adopt it thoroughly.

We thank the reviewer and have defined all abbreviations in the abstract as suggested.

Likewise, in the Text, for example, PDH is not defined as acronym of pyruvate dehydrogenase.

Since PDH was defined in the abstract we did not think to define it again in the text but have now added this as suggested.

MATERIALS & METHODS

Space between paragraphs is missing in Page 2. The title Fatty Acid Synthesis Assay should be closer to the paragraph it refers to (Page 3).

We thank the reviewer and made the changes as suggested.

Paragraph “Cell culture, glucose uptake, and viabilities” – In the title, change “viabilities” to “cell viability”. In the text, mention to HL60 and RHras cells is missing (I believe that the latter should be written RHras rather than RHRAS).

We thank the reviewer and have made the changes as suggested. RHRAS was defined by the creator of the cell line and used all caps to name it.

Lines 73 and 74 – Change “media” to “medium”.

Changed as requested.

Line 102 – The title “Addback Assay” is kind of awkward.

We thank the reviewer and changed the title to Amino Acid Supplementation Assay.

Line 104 – Delete “added either” and add “were added” right before the full stop.

Changed as requested.

RESULTS

Lines 143-144 – The concepts of “mitochondrial stress” and “mitochondrial turnover” should be probably linked to each other.

Changed as requested.

Line 144 – Change “could destabilized” to “could destabilize”.

Changed as requested.

Lines 145-148 – Apparently, HKII was suppressed by both transcriptional and post-translational mechanisms in devimistat-treated cells, while by post-translational mechanisms only in PDHA-deleted cells (however, it appears that bortezomib-induced post-translational suppression accounts for most of the observed effect). The transcriptional effects of devimastat are also shown, for example, in Figure 2A. The Authors should address this point.

We thank the reviewer and have more explicitly stated we are assessing HKII expression levels by QPCR and address the difference between the acute PDH inhibition with devimistat and the chronic loss of activity with the PDHA-deleted cells.

Figure 1G – It should be specified whether cells are treated with devimastat or not.

Figure 1G uses the control and PDHA-deleted cells without the addition of devimistat. This has been added to the text.

Line 156 – The paragraph title should be placed immediately before line 172.

Changed as requested.

Line 178 – Change “manor” to “manner”.

Changed as requested.

Figure 2A - Change “PCK2” to “MFL2” (by analogy with Fig. 2B and 2C) and add “PCK2” to “Relative Expression” (ordinate).

Changed as requested.

Figure 2C – Change “Relative Ratio of PCK2/Actin” to “PCK2/Actin Ratio”.

Changed as requested.

Line 180 – Add a comma after “resistance”.

Changed as requested.

Line 180 – “several AML cell lines” seems actually to refer to several subclones of genetically manipulated MFL2 cells. The point should be clarified here as well as in Figure 2D and in the legend to Figure 2D.

We apologize for the confusion, we have adjusted the text to specify that it is MFL2 cells being treated in figure 2D. The legend for 2D reflects this as well.

Lines 181 and 267– Change “CPI-613” to “devimistat”.  

Changed as requested.

Line 183 – Change “mRNA of the gluconeogenic enzyme, FBP1” to “the mRNA of the gluconeogenic enzyme FBP1” (add “the”, remove comma).

Changed as requested.

Line 195 – Change “the nutrient” to “nutrient”.

Changed as requested.

Line 196 – Resistance: to what should be specified. Probably the two periods of Lines 195-197 should be combined. 

Changed as requested.

Lines 198-200 - Probably the two periods should be combined to state that “Glutamine or asparagine rescued the devimistat-induced reduction of MFL2 and RHRAS cell viability, although the combination of the two aminoacids did not increase the level of rescue (Figure 3A,B)”.

Changed as requested.

Figure 3C,D - Here again, like in the case of Figure 2A-C, panel headings are misleading: Rosa vs PDH k/o cells are also MFL2 cells. Furthermore, in Figure 3D, the indications relative to the abscissa are incomplete, while too many colours are used for the bars. Colours should be four, maximum. I would suggest: Control, Asp (same colour, 2 different concentrations), Dev alone, Dev + Asp (same colour, 2 different concentrations of Asp).

Changed as requested.

Line 219 – Add a comma after “gluconeogenesis”.

Changed as requested.

Line 220 – Change “acetyl-CoA an” to “acetyl-CoA, a”.

Changed as requested.

Line 222 – Change “acetyl-CoA for use in the mitochondria or it can be converted” to “acetyl-CoA, which can be metabolized in the mitochondria or converted”.

Changed as requested.

Line 225 – Change “Carnitine” to “carnitine”.

Changed as requested.

Line 227 – Change “end” to “or”. Line 228 – Change “significant increased” to “significantly increased” or “significant increase of”.

Changed as requested.

Line 229 – Change “reliance of” to “reliance on” or “reliance of cells on”.

Changed as requested.

Line 231 – Change “and was” to “and found”.

Changed as requested.

Line 233 – Change “mitochondrial derived” to “mitochondrion-derived” or “mitochondria-derived”.

Changed as requested.

Line 242 – Change “acetylation however” to “acetylation, while” or “acetylation, whereas” or “acetylation, but”

Changed as requested.

Line 248 – Add commas after “pools” and after “scarcity”.

Changed as requested.

Line 251 – Add comma after “acetylation”.

Changed as requested.

Line 256 – Add comma after “devimistat”. Change “it being” to “the possibility that ACSS be”.

Changed as requested.

Line 258 – Change “inhibition including” to “inhibition; these alterations include”.

Changed as requested.

Line 259 – Change “A summary of the” to “The”.

Changed as requested.

Line 260 – Change “is” to “are”. 

Changed as requested.

Figure 4B – The indications “Vehicle / 50 mM Dev” should be closer to the graph of Figure 4B than to that of Figure 4D.

Figure 4C – This sub-figure is the only one without a title; I would suggest “H3L27 acetylation”. Legend to Figure 5 – “See text for the other abbreviations” should be added at the end of the legend.

DISCUSSION

Line 284 – Change “to supplement needed” to “to supply the needed”.

Changed as requested.

Line 285 – Add “indeed” after “is”.

Changed as requested.

Line 299 – Add a comma after “true”.

Changed as requested.

Line 310 – Add a comma after “acetyl-CoA”.

Changed as requested.

Lines 311-312 – Change “acetyl-CoA levels as it” to “the levels of acetyl-CoA, which”.

Changed as requested.

Line 313 – See comment to Line 229.

Changed as requested.

Line 314 – Change “as” to “is”.

Changed as requested.

Lines 315 and 321 – Add a comma after “devimistat”.

Changed as requested.

Line 323 – Add a comma after “acetylation”.

Changed as requested.

Line 325 – Add a comma after “cells”.

Changed as requested.

Line 330 – Change “to alternative” to “to rely on alternative”.

For clarity this has been left as originally written.

REFERENCES

Reference numbers are repeated twice

Repeated reference numbers maybe an issue with the way the manuscript was formatted, will discuss with the assistant editor to see how to resolve this issue.

We thank the reviewers for their thorough review of our work and believe they have significantly improved the manuscript.